# Masked Two-channel Decoupling Framework for Incomplete Multi-view Weak Multi-label Learning

**Chengliang Liu[1], Jie Wen[1]\*, Yabo Liu[1], Chao Huang[2], Zhihao Wu[1],**
**Xiaoling Luo[3], Yong Xu[1]\***

[1]School of Computer Science and Technology, Harbin Institute of Technology, Shenzhen
[2]School of Cyber Science and Technology, Sun Yat-sen University (Shenzhen Campus)
[3]College of Computer Science and Software Engineering, Shenzhen University
{liucl1996, horatio_ng}@163.com, {jiewen_pr, huangchao_08}@126.com,
yaboliu.ug@gmail.com, xiaolingluoo@outlook.com, yongxu@ymail.com

## Abstract

Multi-view learning has become a popular research topic in recent years, but research on the cross-application of classic multi-label classification and multi-view learning is still in its early stages. In this paper, we focus on the complex yet highly realistic task of incomplete multi-view weak multi-label learning and propose a masked two-channel decoupling framework based on deep neural networks to solve this problem. The core innovation of our method lies in decoupling the single-channel view-level representation, which is common in deep multi-view learning methods, into a shared representation and a view-proprietary representation. We also design a cross-channel contrastive loss to enhance the semantic property of the two channels. Additionally, we exploit supervised information to design a label-guided graph regularization loss, helping the extracted embedding features preserve the geometric structure among samples. Inspired by the success of masking mechanisms in image and text analysis, we develop a random fragment masking strategy for vector features to improve the learning ability of encoders. Finally, it is important to emphasize that our model is fully adaptable to arbitrary view and label absences while also performing well on the ideal full data. We have conducted sufficient and convincing experiments to confirm the effectiveness and advancement of our model.

## 1 Introduction

With the increasing popularity of multi-view learning, researchers have discovered that heterogeneous multi-source features obtained from multiple sensors or techniques can provide a richer and more diverse description of samples [1–4]. For example, features can be extracted using scale-invariant feature transform (SIFT) operator and deep neural networks from different perspectives of an image, or features can be extracted from multimedia data, such as images and texts, to express higher-level semantic objects [5–7]. Many multi-view learning methods are based on the assumptions of multi-view semantic consistency and feature complementarity [8, 9]. They aim to obtain unique sample labels from multi-source features to maintain the unity of multiple views at the semantic level and to leverage the unique advantages of each view to compensate for the limitations of single-view observation [10–12]. For instance, Nie et al. emphasized the importance of learning a cross-view similarity matrix in their work [13]. Li et al. proposed to learn a common graph in the spectral embedding space for final clustering [14]. Han et al. attempted to compute confidence scores of samples on each view to facilitate classification predictions across multiple views [15]. Lin et

---

*Corresponding author

al. proposed dual contrastive prediction for incomplete multi-view representation learning, which performs contrastive learning at both the instance level and category level to ensure cross-view consistency and recover missing information [16]. However, the instance-level contrastive loss of this method only aggregates cross-view representations in the potential space from the perspective of consistency, ignoring the view-private information.

On the other hand, as a classical classification problem, multi-label classification has occupied an important area of pattern recognition for a long time [17]. However, under the upsurge of multi-view learning, new composite problem, i.e., multi-view multi-label classification (MvMLC), are attracting increasing attentions from researchers [18, 19]. As the name suggests, MvMLC can be seen as a supervised subtask within the scope of multi-view learning and can also be considered as an extension of traditional single-view multi-label classification with respect to feature diversity. As a result, existing MvMLC methods not only take into account the basic characteristics of multi-view learning but also consider the requirements of downstream multi-label classification [20, 18, 21]. For example, Zhao et al. proposed a framework called CDMM that mines multi-view consistency and diversity information, which uses the Hilbert-Schmidt Independence Criterion [22] to enhance the diversity of each view and obtain consistent classification results through late fusion [18]. Another advanced non-aligned MvMLC model proposed by Zhao et al. is LVSL (learn view-specific labels), which not only considers the local geometric structure in the original feature space but also introduces a learnable label correlation matrix with low-rank structure [23].

However, these ideal MvMLC methods ignore the possibility of missing multi-view features and labels, making it difficult to achieve good performance on incomplete multi-view weak multi-label data, in which sample's partial views and tags are unknown [24, 25]. In recent years, researchers have proposed some incomplete multi-view weak multi-label classification (iMvWMLC) frameworks to address the adverse effects caused by missing views and labels. For instance, Tan et al. proposed an incomplete multi-view weak label learning model (iMVWL) that decomposes multi-view features into a latent common representation and multiple view-specific basis matrices, and connects the latent representation with labels through a mapping matrix [26]. Additionally, two prior missing indicator matrices are introduced to skip the missing views and labels. Recently, Li et al. developed a non-aligned iMvWMLC model based on traditional matrix factorization, named NAIM3L, to align multiple views in the label space and impose global high-rank and local low-rank constraints on the predicted multi-label matrix [25]. For the double-missing problem, Li et al. introduced a composite indicator matrix for NAIM3L, containing view and label missing information. Another representative deep iMvWMLC framework is DICNet proposed by Liu et al., which applies the contrastive learning technique to aggregate cross-view instances of the same sample and separate instances belonging to different samples [20]. Although DICNet has achieved remarkable improvement, there is a non-negligible category collision problem, i.e., instances belonging to similar samples will be forcibly separated due to this contrastive learning mechanism.

To tackle these issues, we present the **M**asked **T**wo-channel **D**ecoupling framework (**MTD** for short), capable of handling cases where partial views and labels are both missing. The motivation behind this framework is that we both want the underlying representations of the same sample in multiple views to be consistent, while also wanting each view to display its own characteristics, creating a difficult balancing act [10]. This is especially challenging when each view of the same sample is represented by only one feature vector. To address this, we explicitly decouple each view's latent feature into two types of features, i.e., *shared* feature and *view-proprietary* feature, via two-channel encoders. We also propose a novel cross-channel contrastive loss that narrows the distance between positive pairs (shared instances in different views of the same sample) and expands the distance between negative pairs (any cross-view and channel instance pairs of a sample, except positive pairs). Note that our cross-channel contrastive loss does not involve the comparison among samples, so it can well avoid the above class conflict problem and improve the framework's feature decoupling ability. Additionally, we draw inspiration from famous image patch masks [27] and introduce random mask fragments in the input data to guide the encoder's learning of valuable information. Lastly, we design a weak label-guided graph regularization loss to preserve the geometric structure in the embedding space, accounting for the non-equidistance among samples in label space. In summary, our paper makes three key contributions:

1. We propose a novel masked two-channel decoupling framework (MTD) for the iMvWMLC task, which extracts shared and private features for each view separately and enforces this

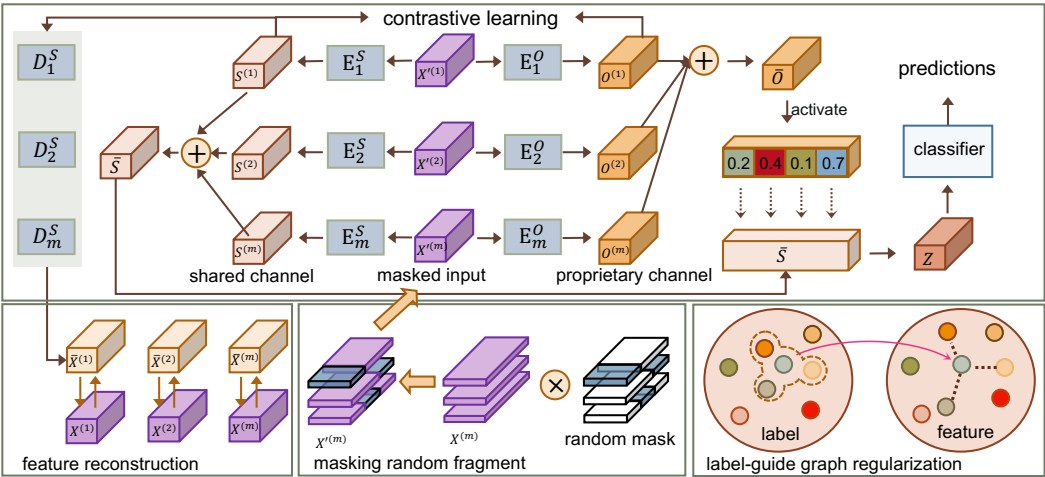

Figure 1: Main architecture of our MTD. Masked data $\{\mathbf{X}'^{(v)}\}_{v=1}^{m}$ is real input of the MTD. And the cross-channel contrastive loss aims to enhance the semantic of 'shared-proprietary' channels.

design with a cross-channel contrastive loss. At the same time, this framework is compatible with arbitrary incomplete multi-view and weak multi-label data.

2. To our best knowledge, we are the first to apply random fragment mask in the field of multi-view learning and achieve significant performance gains, which supports a new multi-view vector data enhancement mechanism for the communication.

3. Different from existing graph regularization based approaches, we utilize supervised label information to build a more reliable topological graph, inducing the embedding features extracted by the encoders to preserve the geometric structure among samples.

## 2 Method

In this section, we comprehensively introduce our model from the following four aspects, namely, two-channel decoupling framework, weak label-guided graph regularization, random fragment masking mechanism, and multi-label classification. First, for the convenience of description, we briefly introduce the formal problem definition and common notations.

### 2.1 Problem definition

In this section, we define the iMvWMLC task as follows: Given multi-view multi-label data $\{\{\mathbf{X}^{(v)}\}_{v=1}^{m}, \mathbf{Y}\}_{v=1}^{(m)}$, where $\mathbf{X}^{(v)}$ is $v$-th view's feature matrix containing $n$ samples and $m$ is the number of views. The $i$-th row of $\mathbf{X}^{(v)} \in \mathbb{R}^{n \times d_v}$ denotes the instance of $i$-th sample on $v$-th view with dimension $d_v$. $\mathbf{Y} \in \{0, 1\}^{n \times c}$ is the label matrix with $c$ categories and $\mathbf{Y}_{i,j} = 1$ denotes the $i$-th sample is tagged as $j$-th class. For missing views, we introduce $\mathbf{W} \in \{0, 1\}^{n \times m}$ as the missing-view index matrix, where $\mathbf{W}_{i,j} = 1$ represent $j$-th view of $i$-th sample is available, otherwise $\mathbf{W}_{i,j} = 0$. For missing labels, we let $\mathbf{G} \in \{0, 1\}^{n \times c}$ be the missing-label index matrix, whose element $\mathbf{G}_{i,j} = 1$ means $i$-th sample's $j$-th label is known, otherwise $\mathbf{G}_{i,j} = 0$. Finally, we fill the unavailable views and the unknown labels with '0' in the data pre-processing step to get final incomplete multi-view feature $\{\mathbf{X}^{(v)}\}_{v=1}^{m}$ and weak label matrix $\mathbf{Y}$. Our goal is to train a neural network that can perform multi-label classification reasoning on incomplete multi-view data. Subscript access $\mathbf{B}_{i,j}$, $\mathbf{B}_{i,:}$, and $\mathbf{B}_{:,j}$ mean the element, row, and column of any matrix $\mathbf{B}$.

### 2.2 Two-channel decoupling framework

As we know, due to the diversity of multi-view data acquisition methods, each view of the original data may have different feature dimensions, which is not conducive to the parallel execution of deep networks. To this end, we map the heterogeneous raw data into a unified embedding space with dimension $d_e$ through stacked encoders. Different from conventional deep multi-view networks, we

employ two groups of multi-layer perceptrons as two channels, shared channel and view-private channel, to extract shared information and view-proprietary information from raw data, respectively, i.e., $\{E_v^S : \mathbf{X}'^{(v)} \rightarrow \mathbf{S}^{(v)}\}_{v=1}^m$ and $\{E_v^O : \mathbf{X}'^{(v)} \rightarrow \mathbf{O}^{(v)}\}_{v=1}^m$. $E_v^S$ and $E_v^O$ denote the shared feature encoder and view-proprietary feature encoder of view $v$, respectively. $\mathbf{X}'^{(v)}$ represents the masked input (See Section 2.4). $\mathbf{S}^{(v)} \in \mathbb{R}^{n \times d_e}$ and $\mathbf{O}^{(v)} \in \mathbb{R}^{n \times d_e}$ are the extracted multi-view shared feature matrix and view-proprietary feature matrix of $n$ samples. The architectures of these two channels are the same but serve different purposes. Specifically, $E_v^S$ tries to mine common features across views, which preserve the sample's basic properties held by all views, while $E_v^O$ focuses on extracting features unique to each view. This design decomposes the discriminative information of each view into two parts, which can satisfy the multi-view consistency and complementarity assumptions at the same time. Of course, it is difficult to achieve the above purposes with only two encoding channels due to the lack of guidance, so we propose a multi-view cross-channel contrastive loss $\mathcal{L}_{ccc}$ to guide the separation of these two kinds of features as follows:

$$
\mathcal{L}_{ccc} = \sum_{i=1}^n \frac{\frac{1}{3N^2 - N}\left[2\sum_{u=1}^m \sum_{v=1}^m \mathbb{1}_{[\Upsilon]}\mathcal{S}\left(s_i^{(u)}, o_i^{(v)}\right)^2 + \sum_{u=1}^m \sum_{v \neq u}^m \mathbb{1}_{[\Upsilon]}\mathcal{S}\left(o_i^{(u)}, o_i^{(v)}\right)^2\right]}{\frac{1}{N^2 - N}\sum_u \sum_{v=u} \mathbb{1}_{[\Upsilon]}\left[\left(\mathcal{S}\left(s_i^{(u)}, s_i^{(v)}\right) + 1\right)/2\right]}, \tag{1}
$$

where $\mathcal{S}$ is the similarity measure function: $\mathcal{S}(x, y) = \frac{x^T y}{\|x\|_2 \cdot \|y\|_2}$ and $\mathbb{1}_{[\Upsilon]}$ is the condition function: $\mathbb{1}_{[\Upsilon]} = 1$ if condition $\{\Upsilon : \mathbf{W}_{i,u}\mathbf{W}_{i,v} = 1\}$ is true, 0 otherwise. $N = \sum_{u,v} \mathbf{W}_{i,u}\mathbf{W}_{i,v}$ denotes the number of valid instance pairs. That is to say we only consider instance pairs that are not missing on both sides. $s_i^{(v)}$ and $o_i^{(v)}$ are features corresponding to $i$-th sample in the shared feature $\mathbf{S}^{(v)}$ and view-proprietary feature $\mathbf{O}^{(v)}$, respectively. According to Eq. (1), our cross-channel contrastive loss consists of two parts, namely, the numerator is the mean of negative pairs' similarities and the denominator is that of positive pairs' similarities. To be more specific, we pair instances from $2N$ channels together, where positive pairs consist of shared features from different views, and the remaining pairs are considered negative. We aim to maximize the similarity between the positive pairs of shared features and minimize the similarity between the negative pairs. This approach fulfills the design objective of the dual-channel model, which is to encourage consistency between shared features from different views while maintaining a clear distinction between each view's view-proprietary feature and other shared or view-proprietary features.

To further enhance the feature extraction capability of the encoders, we introduce stacked decoders to restore the embedding features extracted by the two-channel encoders to original feature space, i.e., $\{D_v : \mathbf{S}^{(v)} \in \mathbb{R}^{n \times d_e} \rightarrow \bar{\mathbf{X}}^{(v)} \in \mathbb{R}^{n \times d_v}\}_{v=1}^m$, where $D_v$ is the decoder corresponding to $v$-th view. $\bar{\mathbf{X}}^{(v)}$ is the reconstructed feature. Finally, we measure the reconstruction quality using the weighted mean square error loss:

$$
\mathcal{L}_{re} = \frac{1}{n}\sum_{v=1}^m \sum_{i=1}^n \frac{1}{d_v}\left\|\bar{\mathbf{X}}_{i,:}^{(v)} - \mathbf{X}_{i,:}^{(v)}\right\|_2^2 \mathbf{W}_{i,v}, \tag{2}
$$

where $\mathbf{W}$ is introduced to mask the unavailable instances. With the $\mathbf{S}^{(v)}$ and $\mathbf{O}^{(v)}$ on each view, we can naturally perform cross-view fusion to obtain the unique shared representation $\bar{\mathbf{S}}$ and $\bar{\mathbf{O}}$ of all samples [28, 20]:

$$
\bar{\mathbf{S}}_{i,:} = \sum_{v=1}^m \frac{\mathbf{S}_{i,:}^{(v)}\mathbf{W}_{i,v}}{\sum_v \mathbf{W}_{i,v}}, \bar{\mathbf{O}}_{i,:} = \sum_{v=1}^m \frac{\mathbf{O}_{i,:}^{(v)}\mathbf{W}_{i,v}}{\sum_v \mathbf{W}_{i,v}}. \tag{3}
$$

Successively, we try to combine shared information and proprietary information to learn consistent representations of samples. Here, unlike the commonly used addition or connection operations, we adopt a novel feature interaction approach, i.e., enhancing the shared information via the proprietary information:

$$
\mathbf{Z}_{i,j} = \theta(\bar{\mathbf{O}}_{i,j}) \cdot \bar{\mathbf{S}}_{i,j}, \tag{4}
$$

where $\theta$ is sigmoid activation function and $\mathbf{Z} \in \mathbb{R}^{n \times d_e}$ is the final fusion representation.

## 2.3 Weak label-guided graph regularization

Many unsupervised multi-view learning methods apply graph regularization technique to maintain the intrinsic structure of data, based on the assumption that two similar samples in the original feature

space should also be similar in the latent space, which has shown good performance in unsupervised scenarios [29–31]. When transferring this assumption to multi-label classification tasks with the characteristics of supervised tasks, we can get a new assumption: two similar samples in the label space should also be similar in the embedding space. Based on this, we attempt to guide the feature extraction process of the encoders by imposing a graph constraint with the supervised information. First, we calculate similarity matrix according to the weak label matrix $\mathbf{Y}$:

$$\mathbf{T}_{i,j} = \frac{\mathbf{C}_{i,j} \cdot (y_i y_j^T)}{\mathbf{C}_{i,j} \cdot (y_i y_j^T) + \eta},$$ 
(5)

where $\mathbf{T} \in [0,1]^{n \times n}$ is the sample similarity graph, describing the similarity between any two samples. $\mathbf{C}_{i,j} = \mathbf{G}_{i,:} \mathbf{G}_{j,:}^T$ is the number of labels available for both samples $i$ and $j$. $y_i$ and $y_j$ denote the $i$-th and $j$-th rows of $\mathbf{Y}$. And $\eta$ is a constant, empirically set to 100 for simplicity. Taking into account the fact that $C_{ij}$ is much larger than $y_i y_j^T$ on most datasets, when the number of categories is large, even if two samples only have one same label, it means that they are much more similar than other sample pairs.

With such a similarity graph, we can constrain the distance between any two samples in the embedding feature space to achieve the purpose of structure preservation by following loss:

$$\mathcal{L}_{gc} = \frac{1}{n^2} \sum_{i=1}^{n} \sum_{j=1}^{n} \left\| \mathbf{Z}_{i,:} - \mathbf{Z}_{j,:} \right\|_2^2 \mathbf{T}_{i,j}.$$ 
(6)

In view of Eq. (6), the more similar two samples are, the greater the penalty weight they are imposed. In order to improve the computational efficiency of the model in the GPU, we rewrite Eq. (6) in the form of matrix product:

$$\mathcal{L}_{gc} = \frac{1}{n^2} Tr(\mathbf{Z}^T \mathbf{L} \mathbf{Z}),$$ 
(7)

where $Tr(\cdot)$ is the trace operation. $\mathbf{L}$ denotes the Laplacian matrix calculated by $\mathbf{L} = \mathbf{D} - \mathbf{T}$, where $\mathbf{D}$ is a diagonal matrix with diagonal element $\mathbf{D}_{i.i} = \sum_{j=1}^{n} \mathbf{T}_{i,j}$.

## 2.4 Masking random fragment of feature

As we all know, the original data contains massive redundant information, which inevitably interferes with the feature extraction process of the model. Inspired by the well-known masked autoencoders (MAE) [27] that randomly mask image patches, we attempt to apply a masking mechanism to the input multi-view vector data. To this end, for any view $v$, we initialize a matrix $\mathbf{M}^{(v)} \in \{0,1\}^{n \times d_v}$ whose elements are all 1 at the same size as the raw data $\mathbf{X}^{(v)}$, and randomly sample $n$ integers $[b_1, b_2, ..., b_n]$ from the range of $[1, d_v - l]$ as the mask starting point of each instance in view $v$, where $l$ is the mask length. Then the $b_i$-th to $(b_i + l)$-th bits of feature $\mathbf{M}_{i,:}^{(v)} \in \mathbb{R}^{d_v}$ are set to 0 to get the final mask matrix $\mathbf{M}^{(v)}$. This matrix stores consecutive mask segments starting at random, enabling local deactivation of feature vectors. We use $m$ mask matrices to mask the original input data, and get corresponding masked feature matrices as the input of the MTD:

$$\{\mathbf{X}'^{(v)}\}_{v=1}^{m} = \{\mathbf{X}^{(v)} \odot \mathbf{M}^{(v)}\}_{v=1}^{m},$$ 
(8)

where $\odot$ means the element-wise multiplication.

At first glance, this is similar to MAE masking random image patches to encourage the model to learn useful features. The difference is that our MTD only partially masks the input features but does not do anything to the encoders, i.e., the encoders still works as if processing the unmasked data. In addition, in our model, we uniformly set the mask rate $\sigma = l/d_v$ to 0.25 for all datasets instead of a larger masking rate like MAE, because we notice the difference that the information density of vector data is much larger than that of image data. Extensive experiments are performed in Section 3.5 to verify the effectiveness of this masking approach.

## 2.5 Weighted multi-label classification and overall loss function

With the fused embedding representation $\mathbf{Z}$, a fully connected layer is employed as a classifier to map $\mathbf{Z}$ to the label space to get the prediction result:

$$\mathbf{P} = \theta(\mathbf{Z} \mathcal{W} + \mathcal{B}),$$ 
(9)

---

**Algorithm 1** Training process of **MTD**

---

**Input:** Incomplete multi-view data $\left\{\mathbf{X}^{(v)}\right\}_{v=1}^{m}$, missing-view index matrix $\mathbf{W}$, weak label $\mathbf{Y}$, missing-label index matrix $\mathbf{G}$.

**Output:** Prediction results $\mathbf{P}$.

1: Initialize model parameters and set hyperparameters ($\alpha$, $\beta$, $\gamma$, learning rate, and training epochs $e$).
2: t=0.
3: **while** $t < e$ **do**
4:      Construct random feature mask matrices $\{\mathbf{M}^{(v)}\}_{v=1}^{m}$.
5:      Compute masked input data $\{\mathbf{X}'^{(v)}\}_{v=1}^{m}$ by Eq. (8).
6:      Extract shared embedding feature $\{\mathbf{S}^{(v)}\}_{v=1}^{m}$ and view-private embedding feature $\{\mathbf{O}^{(v)}\}_{v=1}^{m}$ by two-channel encoders $\{\mathrm{E}_v^S\}_{v=1}^{m}$ and $\{\mathrm{E}_v^O\}_{v=1}^{m}$, respectively.
7:      Compute fused shared embedding feature $\bar{\mathbf{S}}$ and fused view-proprietary feature $\bar{\mathbf{O}}$ by Eq. (3).
8:      Compute final fused representation $\mathbf{Z}$ by Eq. (4).
9:      Compute similarity graph $\mathbf{T}$ by Eq. (5) and corresponding Laplacian matrix $\mathbf{L}$.
10:     Compute cross-channel contrastive loss $\mathcal{L}_{ccc}$ by Eq. (1), reconstruction loss $\mathcal{L}_{re}$ by Eq. (2), and graph embedding loss $\mathcal{L}_{gc}$ by Eq. (7).
11:     Obtain predictions $\mathbf{P}$ by Eq. (9) and compute classification loss $\mathcal{L}_{mc}$ by Eq. (10).
12:     Compute total loss $\mathcal{L}_{all}$ by Eq. (11).
13:     Update network parameters.
14:     $t = t + 1$.
15: **end while**

---

where $\mathcal{W} \in \mathbb{R}^{2d_e \times c}$ and $\mathcal{B} \in \mathbb{R}^{n \times c}$ are learnable parameters of classifier and $\mathbf{P} \in \mathbb{R}^{n \times c}$ is our predicted score matrix. In single-label classification tasks, the cross-entropy loss is usually adopted to guide the learning of model, while in multi-label classification tasks, the prediction of each category is regarded as an independent binary classification problem. Besides, considering the unknown labels in label matrix, we use following multi-label cross-entropy loss as our main classification loss:

$$\mathcal{L}_{mc} = -\frac{1}{nc} \sum_{i=1}^{n} \sum_{j=1}^{c} \left(\mathbf{Y}_{i,j} \log\left(\mathbf{P}_{i,j}\right) + (1 - \mathbf{Y}_{i,j}) \log(1 - \mathbf{P}_{i,j})\right) \mathbf{G}_{i,j}, \quad (10)$$

where $\mathbf{G}$ is introduced to mask the unknown label in the calculation of $\mathcal{L}_{mc}$.

Combined with cross-channel contrastive loss Eq. (1), reconstructed loss Eq. (2), graph embedding loss Eq. (7), and weighted multi-label classification loss Eq. (10), our total loss function can be expressed as:

$$\mathcal{L}_{all} = \mathcal{L}_{mc} + \alpha \mathcal{L}_{gc} + \beta \mathcal{L}_{ccc} + \gamma \mathcal{L}_{re}, \quad (11)$$

where $\alpha$, $\beta$, and $\gamma$ are corresponding penalty parameters. We show the detailed training process in algorithm 1.

## 3 Experiments

### 3.1 Datasets and metrics

Consistent with existing iMvWMLC methods [26, 25, 20], we adopt five famous multi-view multi-label datasets[2] to validate our model, i.e., **Corel5k** [32], **Pascal07** [33], **ESPGame** [34], **IAPRTC12** [35], and **MIRFLICKR** [36]. Six kinds of features are extracted from these datasets as their six views, namely GIST, HSV, Hue, Sift, RGB, and LAB feature. The number of samples in these datasets ranges from 4999 to 25000, and the number of categories ranges from 20 to 291. Please refer to the appendix for more detailed information about the five datasets.

Following plenty of existing works [26, 25, 23], six metrics are selected to evaluate all comparison methods, i.e., Ranking Loss (**RL**), Average Precision (**AP**), Hamming Loss (**HL**), adapted area under curve (**AUC**), OneError (**OE**), and Coverage (**Cov**). To compare performance more intuitively, we actually report results with respect to **1-HL**, **1-RL**, **1-OE**, and **1-Cov**, so for all metrics, higher values indicate better performance.

---

[2] http://lear.inrialpes.fr/people/guillaumin/data.php

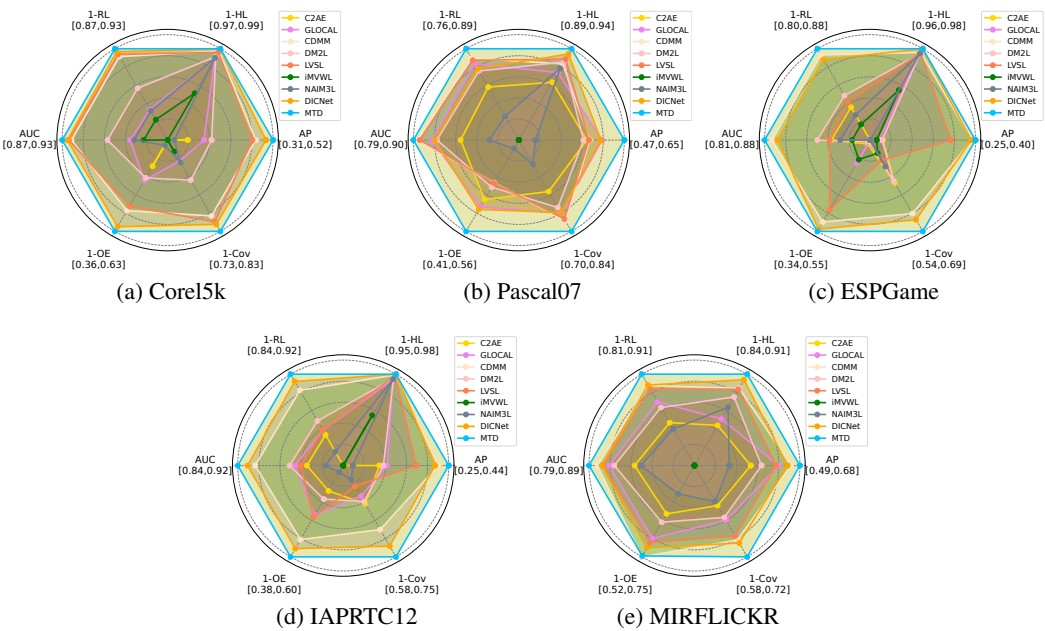

Figure 2: Experimental results of nine methods on the five full databases without any missing views or labels. The center of the radar map shows the worst results and the vertexes mean the best results on the six metrics.

## 3.2 Incomplete multi-view weak multi-label dataset setting

Analogous to previous works [26, 25, 20], we manually construct incomplete multi-view weak multi-label datasets based on above five complete datasets to evaluate the performance of all methods in the case of missing views and missing labels. Concretely, 50% of instances on each view are randomly selected as unavailable instances, which are replaced by '0' value. And we ensure that there is no invalid sample in the dataset, i.e., each sample holds at least one unmissing view. For the weak labels, we set half of the positive and negative tags for each category to be unknown. 70% of samples with missing views and missing labels are randomly selected as the training set. The construction of incomplete data is repeated many times to reduce the randomness of experiment.

## 3.3 Comparison methods

In order to evaluate the advancement of our framework, eight top comparison methods are selected in our experiment, i.e., **C2AE** [37], **GLOCAL** [38], **CDMM** [18], **DM2L** [39], **LVSL** [23], **NAIM3L** [25], **iMVWL** [26], and **DICNet** [20], to compare with our MTD on the five incomplete multi-view weak multi-label datasets. To date, few MvMLC methods can handle missing views and weak labels at the same time, thus we introduce some related multi-label classification methods in our experiments. To be more specific, of the nine methods, iMVWL, NAIM3L, DICNet, and our MTD are perfectly applicable to the iMvWMLC task. C2AE, GLOCAL, and DM2L are single-view multi-label classification methods that can deal with incomplete labels, so we conduct these methods on each view and record the best result. CDMM and LVSL are the MvMLC methods unable to handle any data missing, so we simply complete the missing views using the mean of available instances and fill the unknown tags with '0'. The statistical information of competitors is shown in the appendix.

## 3.4 Experimental results and analysis

To evaluate the performance of our method in scenarios with missing views and labels, we compare it against eight state-of-the-art algorithms on five datasets. The mean and standard deviation (decimal form in the bottom right) of the results are reported in Table 1. In addition to the six performance metrics mentioned above, we also calculate the average ranking of each algorithm based on these

Table 1: Experimental results of nine methods on the five databases with 50% missing-view rate, 50% missing-label rate, and 70% training samples. 'Ave.R' denotes the average ranking of the corresponding method on all six metrics.

| Data | Metric Sources | C2AE[37] AAAI '17 | GLOCAL[38] TKDE '17 | CDMM[18] KBS '20 | DM2L[39] PR '21 | LVSL[23] TMM '22 | iMVWL[26] IJCAI '18 | NAIM3L[25] TPAMI'22 | DICNet[20] AAAI '23 | MTD – |
|---|---|---|---|---|---|---|---|---|---|---|
| Corel5k | AP | $0.227_{0.008}$ | $0.285_{0.004}$ | $0.354_{0.004}$ | $0.262_{0.005}$ | $0.342_{0.004}$ | $0.283_{0.008}$ | $0.309_{0.004}$ | $0.381_{0.004}$ | $0.415_{0.008}$ |
| | 1-HL | $0.980_{0.002}$ | $0.987_{0.000}$ | $0.987_{0.000}$ | $0.987_{0.000}$ | $0.987_{0.000}$ | $0.978_{0.000}$ | $0.987_{0.000}$ | $0.988_{0.000}$ | $0.988_{0.000}$ |
| | 1-RL | $0.804_{0.010}$ | $0.840_{0.003}$ | $0.884_{0.003}$ | $0.843_{0.002}$ | $0.881_{0.003}$ | $0.865_{0.005}$ | $0.878_{0.002}$ | $0.882_{0.004}$ | $0.893_{0.004}$ |
| | AUC | $0.806_{0.010}$ | $0.843_{0.003}$ | $0.888_{0.003}$ | $0.845_{0.002}$ | $0.884_{0.003}$ | $0.868_{0.005}$ | $0.881_{0.002}$ | $0.884_{0.004}$ | $0.896_{0.004}$ |
| | 1-OE | $0.246_{0.016}$ | $0.327_{0.010}$ | $0.410_{0.007}$ | $0.295_{0.014}$ | $0.391_{0.015}$ | $0.311_{0.015}$ | $0.350_{0.009}$ | $0.468_{0.007}$ | $0.491_{0.012}$ |
| | 1-Cov | $0.596_{0.016}$ | $0.648_{0.006}$ | $0.723_{0.007}$ | $0.647_{0.005}$ | $0.718_{0.006}$ | $0.702_{0.008}$ | $0.725_{0.005}$ | $0.727_{0.011}$ | $0.749_{0.009}$ |
| | Ave.R | 8.83 | 6.33 | 2.83 | 6.83 | 3.83 | 6.83 | 4.33 | 2.17 | **1.00** |
| Pascal07 | AP | $0.485_{0.008}$ | $0.496_{0.004}$ | $0.508_{0.005}$ | $0.471_{0.008}$ | $0.504_{0.005}$ | $0.437_{0.018}$ | $0.488_{0.003}$ | $0.505_{0.012}$ | $0.551_{0.004}$ |
| | 1-HL | $0.908_{0.002}$ | $0.927_{0.000}$ | $0.931_{0.001}$ | $0.928_{0.001}$ | $0.930_{0.000}$ | $0.882_{0.004}$ | $0.928_{0.001}$ | $0.929_{0.001}$ | $0.932_{0.001}$ |
| | 1-RL | $0.745_{0.009}$ | $0.767_{0.004}$ | $0.812_{0.004}$ | $0.761_{0.005}$ | $0.806_{0.003}$ | $0.736_{0.015}$ | $0.783_{0.001}$ | $0.783_{0.008}$ | $0.831_{0.003}$ |
| | AUC | $0.765_{0.010}$ | $0.786_{0.003}$ | $0.838_{0.003}$ | $0.779_{0.004}$ | $0.832_{0.002}$ | $0.767_{0.015}$ | $0.811_{0.001}$ | $0.809_{0.006}$ | $0.851_{0.003}$ |
| | 1-OE | $0.438_{0.008}$ | $0.443_{0.005}$ | $0.419_{0.008}$ | $0.420_{0.011}$ | $0.419_{0.008}$ | $0.362_{0.023}$ | $0.421_{0.006}$ | $0.427_{0.015}$ | $0.459_{0.007}$ |
| | 1-Cov | $0.680_{0.010}$ | $0.703_{0.004}$ | $0.759_{0.003}$ | $0.692_{0.004}$ | $0.751_{0.003}$ | $0.677_{0.015}$ | $0.727_{0.002}$ | $0.731_{0.006}$ | $0.784_{0.003}$ |
| | Ave.R | 7.17 | 5.33 | 2.83 | 6.67 | 3.83 | 8.83 | 4.83 | 4.00 | **1.00** |
| ESPGame | AP | $0.202_{0.006}$ | $0.221_{0.002}$ | $0.289_{0.003}$ | $0.212_{0.002}$ | $0.285_{0.003}$ | $0.244_{0.005}$ | $0.246_{0.002}$ | $0.297_{0.002}$ | $0.306_{0.003}$ |
| | 1-HL | $0.971_{0.002}$ | $0.982_{0.000}$ | $0.983_{0.000}$ | $0.982_{0.000}$ | $0.983_{0.000}$ | $0.972_{0.000}$ | $0.983_{0.000}$ | $0.983_{0.000}$ | $0.983_{0.000}$ |
| | 1-RL | $0.772_{0.006}$ | $0.780_{0.004}$ | $0.832_{0.001}$ | $0.781_{0.001}$ | $0.829_{0.001}$ | $0.808_{0.002}$ | $0.818_{0.002}$ | $0.832_{0.001}$ | $0.837_{0.002}$ |
| | AUC | $0.777_{0.006}$ | $0.784_{0.004}$ | $0.836_{0.001}$ | $0.785_{0.001}$ | $0.833_{0.002}$ | $0.813_{0.002}$ | $0.824_{0.002}$ | $0.836_{0.001}$ | $0.842_{0.002}$ |
| | 1-OE | $0.262_{0.018}$ | $0.317_{0.005}$ | $0.396_{0.005}$ | $0.294_{0.006}$ | $0.389_{0.004}$ | $0.343_{0.013}$ | $0.339_{0.003}$ | $0.439_{0.007}$ | $0.447_{0.009}$ |
| | 1-Cov | $0.497_{0.011}$ | $0.496_{0.006}$ | $0.574_{0.004}$ | $0.488_{0.003}$ | $0.567_{0.005}$ | $0.548_{0.004}$ | $0.571_{0.003}$ | $0.593_{0.003}$ | $0.602_{0.004}$ |
| | Ave.R | 8.67 | 7.33 | 2.33 | 7.50 | 3.67 | 6.17 | 4.33 | 1.83 | **1.00** |
| IAPRTC12 | AP | $0.224_{0.007}$ | $0.256_{0.002}$ | $0.305_{0.004}$ | $0.234_{0.003}$ | $0.304_{0.004}$ | $0.237_{0.003}$ | $0.261_{0.001}$ | $0.323_{0.001}$ | $0.332_{0.003}$ |
| | 1-HL | $0.965_{0.002}$ | $0.980_{0.000}$ | $0.981_{0.000}$ | $0.980_{0.000}$ | $0.981_{0.000}$ | $0.969_{0.000}$ | $0.980_{0.000}$ | $0.981_{0.000}$ | $0.981_{0.000}$ |
| | 1-RL | $0.806_{0.005}$ | $0.825_{0.002}$ | $0.862_{0.002}$ | $0.823_{0.002}$ | $0.861_{0.002}$ | $0.833_{0.002}$ | $0.848_{0.001}$ | $0.873_{0.001}$ | $0.875_{0.002}$ |
| | AUC | $0.807_{0.005}$ | $0.830_{0.001}$ | $0.864_{0.002}$ | $0.825_{0.001}$ | $0.863_{0.001}$ | $0.835_{0.001}$ | $0.850_{0.001}$ | $0.874_{0.000}$ | $0.876_{0.001}$ |
| | 1-OE | $0.300_{0.031}$ | $0.378_{0.007}$ | $0.432_{0.008}$ | $0.340_{0.008}$ | $0.429_{0.009}$ | $0.352_{0.008}$ | $0.390_{0.006}$ | $0.468_{0.002}$ | $0.467_{0.005}$ |
| | 1-Cov | $0.523_{0.009}$ | $0.534_{0.003}$ | $0.597_{0.004}$ | $0.529_{0.004}$ | $0.597_{0.005}$ | $0.564_{0.005}$ | $0.592_{0.004}$ | $0.649_{0.001}$ | $0.649_{0.004}$ |
| | Ave.R | 9.00 | 6.33 | 2.67 | 7.50 | 3.33 | 6.67 | 5.00 | 1.50 | **1.17** |
| MIRFLICKR | AP | $0.505_{0.008}$ | $0.537_{0.002}$ | $0.570_{0.002}$ | $0.514_{0.006}$ | $0.553_{0.002}$ | $0.490_{0.012}$ | $0.551_{0.002}$ | $0.589_{0.005}$ | $0.607_{0.004}$ |
| | 1-HL | $0.853_{0.004}$ | $0.874_{0.001}$ | $0.886_{0.001}$ | $0.878_{0.001}$ | $0.885_{0.001}$ | $0.839_{0.002}$ | $0.882_{0.001}$ | $0.888_{0.002}$ | $0.891_{0.001}$ |
| | 1-RL | $0.821_{0.003}$ | $0.832_{0.001}$ | $0.856_{0.001}$ | $0.831_{0.003}$ | $0.856_{0.001}$ | $0.803_{0.008}$ | $0.844_{0.001}$ | $0.863_{0.004}$ | $0.875_{0.002}$ |
| | AUC | $0.810_{0.004}$ | $0.828_{0.001}$ | $0.846_{0.001}$ | $0.828_{0.003}$ | $0.844_{0.001}$ | $0.787_{0.012}$ | $0.837_{0.001}$ | $0.849_{0.004}$ | $0.862_{0.002}$ |
| | 1-OE | $0.505_{0.020}$ | $0.552_{0.005}$ | $0.631_{0.004}$ | $0.510_{0.008}$ | $0.607_{0.004}$ | $0.511_{0.022}$ | $0.585_{0.003}$ | $0.637_{0.007}$ | $0.655_{0.004}$ |
| | 1-Cov | $0.590_{0.005}$ | $0.605_{0.003}$ | $0.640_{0.001}$ | $0.604_{0.005}$ | $0.636_{0.001}$ | $0.572_{0.013}$ | $0.631_{0.002}$ | $0.652_{0.007}$ | $0.676_{0.004}$ |
| | Ave.R | 8.17 | 6.17 | 3.00 | 6.83 | 3.83 | 8.67 | 5.00 | 2.00 | **1.00** |

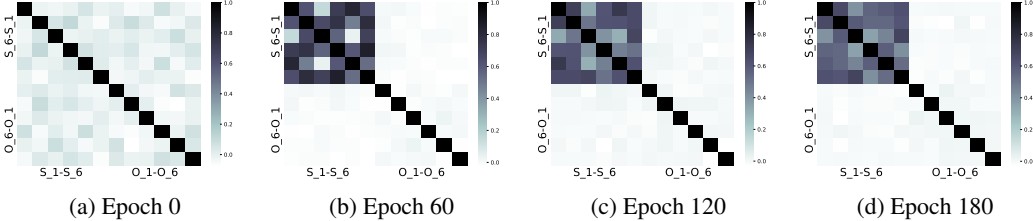

| (a) Epoch 0 | (b) Epoch 60 | (c) Epoch 120 | (d) Epoch 180 |
|---|---|---|---|

Figure 3: A random sample's channel similarity heat maps across all channels on Corel5k dataset with half of missing views and labels. S_1- S_6 and O_1- O_6 denote shared features and view-proprietary features on six views, respectively. With the increase of training epoch, the similarities of features on shared and proprietary channels show the expected trend, that is, the shared features across views gradually converged, while the similarities of "shared-proprietary" and "proprietary-proprietary" feature pairs gradually decreased.

metrics, denoted as 'Ave.R', for intuitive comparison. Based on the results in Table 1, we make the following observations:

1. Our MTD is compelling, outperforming comparison methods on almost all metrics across all five datasets. In addition to the average ranking of 1.17 on the IAPRTC12 dataset, it

ranks 1st on the other four datasets. Whether compared with traditional methods (such as GLOCAL and NAIM3L) or deep methods (such as C2AE and DICNet), our MTD shows excellent compatibility with missing views and weak labels.

2. We can find that methods based on deep neural networks generally perform better than traditional methods. Specifically, our MTD, DICNet, and CDMM are consistently ranked in the top 3 across all five datasets, which can be attributed to the strong feature extraction capabilities of deep encoders and the flexible multi-view fusion strategy they employ.

3. In the comparison among traditional methods, the method that is specifically designed for double incompleteness shows obvious advantages compared to other methods. This indicates the necessity of considering possible missing views and labels during the model design process.

In addition to experimenting on datasets with 50% missing views and missing labels, we also perform comparison experiments on the full dataset without any missing data and use radar charts to show the results in Fig. (2). For more intuitive observation, we set the range for each axis to the min-max values of the nine methods on the corresponding metric. It can be seen from Fig. (2) that the performance of our method on the complete datasets is still outstanding, which verifies the good adaptation of it.

To demonstrate that our cross-channel contrastive loss works as expected, we plot a random sample's channel similarity heat maps across all channels in different training epochs on the Corel5k dataset with half of missing views and labels in Fig. (3). Each patch of the heat map shows the cosine similarity of corresponding two instances, i.e., $i$-th sample's heat map $\mathbf{H}_i = \mathbf{C}\mathbf{C}^T$, where $\mathbf{C} = (s_i^{(1)}, s_i^{(2)}, ..., s_i^{(m)}, o_i^{(1)}, o_i^{(2)}, ..., o_i^{(m)})^T \in \mathbb{R}^{2m \times d_e}$. $s_i^{(1)} \in \mathbb{R}^{d_e}$ and $o_i^{(1)} \in \mathbb{R}^{d_e}$ are the instances on shared and view-proprietary channels of first view of sample $i$, respectively. In Fig. (3), it is evident that in the initial state, the similarities between different channels are relatively small. However, as training progresses, the similarities between instance pairs on the first $m$ shared channels increase rapidly, while the similarities for the remaining channels are close to zero. This observation indicates that our cross-channel contrastive loss is functioning as expected. Additionally, we notice that there is a shared feature on channel 3 that is not well-aggregated. This may be due to the third view of this sample sharing less information with the other views.

## 3.5 Ablation study

Table 2: Ablation results of our MTD on Corel5k dataset with 50% missing views, 50% missing labels. 's_ch' denotes the conventional single-channel network and 'd_ch' denotes our double-channel framework. '*w/o*' means 'without'.

| Method | AP | 1-HL | 1-RL | AUC | 1-OE | 1-Cov |
|---|---|---|---|---|---|---|
| s_ch baseline | 0.396 | 0.988 | 0.888 | 0.891 | 0.469 | 0.738 |
| d_ch baseline | 0.408 | 0.988 | 0.890 | 0.893 | 0.482 | 0.744 |
| d_ch+$\mathcal{L}_{gc}$ | 0.409 | 0.988 | 0.890 | 0.893 | 0.488 | 0.744 |
| d_ch+$\mathcal{L}_{re}$+$\mathcal{L}_{gc}$ | 0.413 | 0.988 | 0.890 | 0.894 | 0.487 | 0.746 |
| **MTD** | 0.415 | 0.988 | 0.893 | 0.896 | 0.491 | 0.750 |
| MTD *w/o* mask | 0.397 | 0.987 | 0.888 | 0.891 | 0.466 | 0.738 |
| s_ch *w/o* mask | 0.372 | 0.987 | 0.881 | 0.884 | 0.449 | 0.721 |
| d_ch *w/o* mask | 0.387 | 0.987 | 0.885 | 0.889 | 0.452 | 0.732 |

Our model contains multiple loss functions and strategies, in order to further study the effectiveness of each component, we perform extensive ablation experiments on the Corel5k dataset with 50% missing labels and views, and report results in Table 2. Specifically, we take the common single-channel model [20, 40] as a baseline architecture and extend it to a two-channel baseline framework. We iteratively add our individual loss functions $\mathcal{L}_{gc}$, $\mathcal{L}_{re}$, and $\mathcal{L}_{ccc}$ into it and remove the masking operation from different versions. From Table 2, we observe that all designs play a role to varying degrees, especially the fragment masking mechanism's performance increase on the Corel5k dataset is the most noticeable. This confirms that the random masking mechanism to raw data, commonly used in the image and text analysis domain, is also effective in the field of multi-view learning. However, this strategy still needs to be further studied for the underlying mechanism of action.

### 3.6 Implementation details

Our TMD is implemented by Pytorch and MindSpore frameworks on Ubuntu operating system with a single RTX 3090 GPU and an i7-12900k CPU. The learning rate is set to 0.1 and the Stochastic Gradient Descent (SGD) optimizer is chosen for training model. The batch size and momentum are 128 and 0.9 for all five datasets. Please refer to the appendix for the setting of other hyperparameters. The code can be found at `https://justsmart.github.io/`.

## 4 Conclusion

In this paper, we propose a novel masked two-channel decoupling framework (MTD) for the iMvWMLC task. Our MTD decouples single-channel features commonly used in previous methods into two-channel features for consensus and complementarity learning, and we design a cross-channel contrastive loss for this setting. Additionally, we employ a random fragment masking strategy during the training phase to reduce redundancy of raw data, inspired by image and text masking strategies, which leads to an obvious performance boost. Furthermore, we introduce a label-guided graph constraint approach to ensure that the learned embedding features maintain structure information among samples. Our extensive comparison and ablation experiments demonstrate that MTD outperforms other advanced methods and is compatible with arbitrary incomplete multi-view and weak multi-label data. Looking back at our work, although we propose many strategies and techniques to advance the frontier research progress of iMvWMLC, the complexity of this issue leaves room for further optimization, such as considering multi-label correlation and missing data recovery as directions of future works.

## Acknowledgements

This work is supported in part by the Shenzhen Higher Education Stability Support Program under Grant No. GXWD20220811173317002, in part by the Chinese Association for Artificial Intelligence (CAAI)-Huawei MindSpore Open Fund under Grant No. CAAIXSJLJJ-2022-011C.

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

# A Datasets

Table 3: Detailed statistics about five multi-view multi-label databases.

| Database | # Sample | # Category | # View | # Category/#Sample |
|----------|----------|------------|--------|--------------------|
| Corel5k | 4999 | 260 | 6 | 3.40 |
| Pascal07 | 9963 | 20 | 6 | 1.47 |
| ESPGame | 20770 | 268 | 6 | 4.69 |
| IAPRTC12 | 19627 | 291 | 6 | 5.72 |
| MIRFLICKR | 25000 | 38 | 6 | 4.72 |

These datasets are uniformly extracted six kinds of feature as six views from five image datasets, i.e., GIST, HSV, Hue, Sift, RGB, and LAB. Table 3 shows the statistical information of the five datasets.

# B Comparison Methods

Table 4: Information of comparison multi-label classification methods. 'Multi-view' means the method is designed for multi-view data; 'Missing-view' and 'Missing-label' represent the corresponding method is able to handle incomplete views and weak labels.

| Method | Sources | Multi-view | Missing-view | Missing-label |
|--------|---------|------------|--------------|---------------|
| C2AE[37] | AAAI '17 | ✗ | ✗ | ✓ |
| GLOCAL[38] | TKDE '17 | ✗ | ✗ | ✓ |
| CDMM[18] | KBS '20 | ✓ | ✗ | ✗ |
| DM2L[39] | PR '21 | ✗ | ✗ | ✓ |
| LVSL[23] | TMM '22 | ✓ | ✗ | ✗ |
| iMVWL[26] | IJCAI '18 | ✓ | ✓ | ✓ |
| NAIM3L[25] | TPAMI '22 | ✓ | ✓ | ✓ |
| DICNet[20] | AAAI '23 | ✓ | ✓ | ✓ |

# C Hyperparameters study

The overall loss of our model mainly contains 3 hyperparameters, i.e., $\alpha$, $\beta$, and $\gamma$. In order to study the impact of these three hyperparameters on the performance of the model, we iteratively set different values and report the corresponding AP values in Fig. 4:

In Fig. 4, we show the AP value with respect to different parameter selections on Corel5k and Pascal07 datasets with 50% missing views and labels. Specifically, from Fig. 4a and 4b, we recommend selecting $\alpha$ from the ranges of $[1e-4, 1e-3]$ and $[1e-4, 1e0]$ for Corel5k and Pascal07 datasets respectively; selecting $\beta$ from the ranges of $[1e-3, 1e-1]$ and $[1e-3, 1e-1]$ for Corel5k and Pascal07 datasets respectively. For convenience, we uniformly set $\alpha = 0.4$ and $\beta = 0.4$ in our experiments. For $\gamma$, our method is not sensitive to it, so we set $\gamma = 0.1$ for all five datasets.

From Fig. 4d and Fig. 4e, it can be seen that too large or too small mask rate is not conducive to achieving optimal performance. Based on experiments, we recommend setting a mask rate from 0.2 to 0.4 (0.25 is used for all datasets in our experiments for convenience).

# D Time cost

To study the training and test efficiency of our MTD, we report the time cost of the training and test phases of the six supervised methods on the Corel5k dataset with 70% training samples in Table 5. The convergence setting of the model plays a critical role in determining the training time required. Therefore, we measure the running times of all methods under their default convergence conditions.

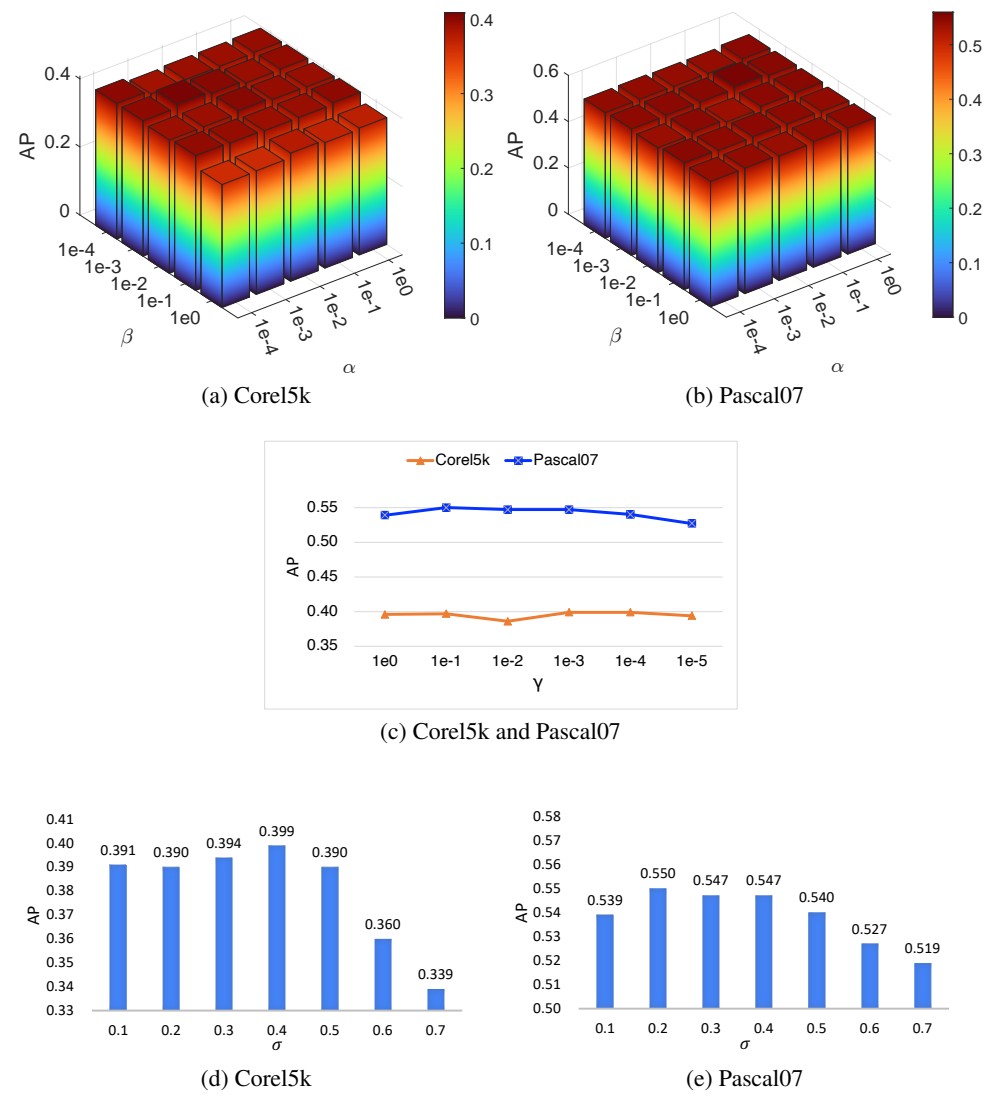

Figure 4: AP value v.s. hyperparameters $\alpha$ and $\beta$ on the (a) Corel5k and (b) Pascal07 datasets; AP value v.s. hyperparameter $\gamma$ on the (c) Corel5k and Pascal07 datasets; AP value v.s. hyperparameter $\sigma$ on the (d) Corel5k and (e) Pascal07 datasets. The two datasets are with 50% missing-label rate, 50% missing-view rate, and 70% training samples.

Table 5: Time efficiency of training and testing phases of the nine methods on the Corel5k dataset with 70% training samples. (Unit: second)

| Phase \ Method | C2AE | GLOCAL | CDMM | DM2L | LVSL | iMVWL | NAIML | DICNet | MTD |
|---|---|---|---|---|---|---|---|---|---|
| Training | 170.24 | 154.42 | 16.02 | 713.37 | 63.73 | 165.82 | 143.63 | 313.89 | 687.52 |
| Test | 0.04 | 0.89 | 1.73 | 0.04 | 0.64 | 0.02 | 0.01 | 0.05 | 0.15 |

For methods that process a single view at a time, we report both the total training time for all views and the test time for a single view. For DICNet and MTD, we report the time spent in 100 epochs. In order to ensure consistent hardware conditions, all tests are carried out on the same computer. From the results shown in Table 5, it can be observed that our deep learning-based method requires longer training times, but reaches higher performance than other methods.

