# OpenReview forum: "Masked Two-channel Decoupling Framework for Incomplete Multi-view Weak Multi-label Learning"
_NeurIPS.cc/2023/Conference — NeurIPS 2023 poster_

### Official Review · Reviewer_XPPT · 2023-06-27

**Soundness:** 3 good
**Presentation:** 4 excellent
**Contribution:** 3 good
**Rating:** 6
**Confidence:** 5

**Summary:**

The core innovation of the method lies in decoupling the single-channel view-level representation, which is common in deep multi-view learning methods, into a shared representation and a view-proprietary representation with a cross-channel contrastive loss. The authors have conducted sufficient experiments to verify the effectiveness of the proposed method.

**Strengths:**

1. The definition of the problem in this paper is clear. The topic is novel, and as far as I know, multi-view multi-label classification methods for the complex situation of missing labels and data are underexplored.

2. The interpretation of extracting multi-view features from two perspectives is intuitive and fits with the fundamental assumption of multi-view/multi-modal learning problems that there are commonalities as well as differences between modalities.

3. The experimental results are sufficient. The authors not only provide the results in the missing case, but also provide the results of the proposed model on the complete datasets, which shows that the model has good learning ability for multi-view data.

**Weaknesses:**

1. In Eq.(1), the necessary explanation about the coefficient 2 in the numerator is lacking. The authors should also provide more explanation about the motivation of such design.

2. A clarification on the value of mask rate $\sigma$ is lacking.

3. The authors propose to build two groups of encoder-decoder for different views, but the computational complexity may increase with the number of views quickly.

4.  There are some unclear expressions, for instance, in Section 2.2, the authors say “Different from conventional deep multi-view networks”. So what are the conventional deep multi-view networks? There is no corresponding reference here.

**Questions:**

Is it efficient to address the missing problem by introducing additional prior information, that is matrix G and W in this paper?

**Limitations:**

 The authors have discussed the limitations and suggested further directions for improvement in the Conclusion.

---

> ### Author Rebuttal · Authors · 2023-08-10
>
> Thank you very much for your recognition and we respond to your comments below:
>
> W1: In Eq.(1), the necessary explanation about the coefficient 2 in the numerator is lacking. The authors should also provide more explanation about the motivation of such design.
>
> A1: Thanks for your comments, we have added more explanation about coefficient 2 in molecular of Eq. (1):
>
> In addition, considering the efficiency of matrix calculation, we set the coefficient of the contrastive item of shared features and private features to 2 in the numerator part, so that the entire paired similarities can be obtained by a single matrix multiplication, i.e., $[s_i^{(1)}; s_i^{(2)};…; s_i^{(N)}; o_i^{(1)}; o_i^{(2)};…; o_i^{(N)}]\times [s_i^{(1)}; s_i^{(2)};…; s_i^{(N)}; o_i^{(1)}; o_i^{(2)};…; o_i^{(N)}]=Q\in \mathbb{R}^{2N\times 2N}$. The matrix $Q$ can be divided into four sub-matrices whose size is $N$ by $N$. The sub-matrix in the upper left corner stores the similarity values of the denominator in Eq. (1), the sub-matrix in the lower right corner stores the values of the second item in the numerator, and the remaining two submatrix, which are transposed to each other, store the similarity values of the first item in the numerator. Therefore, we set the coefficient of the first term in the numerator to 2. At the same time, the diagonal elements of $Q$ are removed, so that the number of elements in the numerator and denominator parts is $\frac{1}{3N^{2}-N}$ and $\frac{1}{N^{2}-N}$ respectively.
>
> W2: A clarification on the value of mask rate $\sigma$ is lacking.
>
> A2: Thanks for your comments, as described in our paper, we empirically set $\sigma$ to 0.25 instead of 0.75 in MAE, considering that vector-based features do not have high redundancy as in image data. At the same time, we give the parameter analysis of $\sigma$ in the supplementary material, and the experimental results show that the model achieves the best performance when $\sigma$ is set to 0.2-0.4.
>
> W3: The authors propose to build two groups of encoder-decoder for different views, but the computational complexity may increase with the number of views quickly.
>
> A3: Thank you for your comments. In fact, we designed only 2 encoders for each view (one decoder per view). Since our codec structure is simple, including only 4-5 linear layers, the increase in model parameters brought about by increasing the number of views is also linear.
>
> W4: There are some unclear expressions, for instance, in Section 2.2, the authors say “Different from conventional deep multi-view networks”. So what are the conventional deep multi-view networks? There is no corresponding reference here.
>
> A4: We are sorry that our expression confused you. We have added relevant references to the manuscript:
>
> Different from conventional deep multi-view networks[1-3], we employ two groups of multi-layer…
>
> [1] Deep Double Incomplete Multi-View Multi-Label Learning With Incomplete Labels and Missing Views, TNNLS
>
> [2] Multi-level Feature Learning for Contrastive Multi-view Clustering, CVPR
>
> [3] Deep Embedded Complementary and Interactive Information for Multi-View Classification, AAAI
>
> Q1: Is it efficient to address the missing problem by introducing additional prior information, that is matrix G and W in this paper?
>
> A1: Thanks for your question, the missing view indicator matrix $\mathbf{W}$ and missing label indicator matrix $\mathbf{G}$ we introduced in the paper can avoid the negative impact of missing information. And our experiments in the new supplementary material confirm that introducing $\mathbf{W}$ and $\mathbf{G}$ can effectively improve the model's ability to handle missing views and labels.

---

> > ### Comment · Reviewer_XPPT · 2023-08-12
> >
> > Thanks for the authors' responses. The authors have addressed my concerns and I'd like to rise my score.

---

> > > ### Author Response · Authors · 2023-08-21
> > > **Response to Reviewer XPPT**
> > >
> > > Thank you very much for your review comments and encouragement, we will further improve the quality of the manuscript.

---

### Official Review · Reviewer_txHg · 2023-06-30

**Soundness:** 4 excellent
**Presentation:** 3 good
**Contribution:** 3 good
**Rating:** 6
**Confidence:** 5

**Summary:**

This paper proposes a general framework for missing multi-view and missing multi-label classification. The main innovation is to  decouple each view feature into two different  shared and private features. The paper also applies random masks to the raw features to make the network learn from limited information, and shows the benefits of this approach with ablation experiments.

**Strengths:**

- This paper is clear in terms of writing and presentation. It is easy for the reader to understand the author's motivation.

- The problem addressed by the authors is complex (doubly incomplete data) and poses a greater challenge than methods based on ideally complete data. However, judging from the results, the author solves this problem well.

- The method of using labels to constrain the feature extraction is novel, and the author's local masking of input features has also brought performance growth. I think there are simple and effective techniques that can be extended to other fields.

- The experimental results show that the MTD has significant advantages over the existing multi-view multi-label classification methods.


**Weaknesses:**

- The one-HL metric does not seem to be of much significance because it does not effectively evaluate the performance of different methods according to the author's experimental results.
- It is inappropriate to appear both ‘Figure’ and ‘Fig.’ in the text.
- The author should clearly explain the meaning of Figure 3 in the caption.


**Questions:**

- Are these masks fixed during training? How exactly did the authors set up the mask?
- Why the results of iMVWL and NAIM3L are not consensus with original papers?

---

> ### Author Rebuttal · Authors · 2023-08-10
>
> Thank you very much for your recognition and we respond to your comments below:
>
> W1: The one-HL metric does not seem to be of much significance because it does not effectively evaluate the performance of different methods according to the author's experimental results.
>
> A1: Yes, the metric "1-HL" does not show strong discrimination in performance evaluation, however, as an old multi-label classification metric, we still include it in the experimental results and to make a comprehensive comparison with existing works (iMvWLL, NAIM3L, and DICNet).
>
> W2: It is inappropriate to appear both ‘Figure’ and ‘Fig.’ in the text.
>
> A2: Thanks for your suggestion, we have checked the full text to ensure the consistency of similar expressions.
>
> W3: The author should clearly explain the meaning of Figure 3 in the caption.
>
> A3: Thanks for your comments, we have added a detailed explanation to the caption of Figure 3 in the manuscript:
>
> A random sample's channel similarity heat maps across all channels on Corel5k dataset with half of missing views and labels. S_1- S_6 and O_1- O_6 denote shared features and view-proprietary features on six views, respectively. With the increase of training epoch, the similarities of features on shared and proprietary channels show the expected trend, that is, the shared features across views gradually converged, while the similarities of "shared-proprietary" and "proprietary-proprietary" feature pairs gradually decreased.
>
> Response to some questions:
>
> Q1: Are these masks fixed during training? How exactly did the authors set up the mask?
>
> A1: Yes, these masks are fixed during training. In our model, we uniformly set the mask rate $\sigma=l/d_v$ to 0.25 for all datasets instead of a larger masking rate like MAE. The reason is that we notice the difference that the information density of vector data is much larger than that of image data. At the same time, we give the parameter analysis of $\sigma$ in the supplementary material, and the experimental results show that the model achieves the best performance when $\sigma$ is set to 0.2-0.4.
>
> Q2: Why the results of iMVWL and NAIM3L are not consensus with original papers?
>
> A2: Since we added new metrics (1-OE and 1-Cov) to our experiment that were not provided in the original papers, we reproduced their methods on our incomplete data and our replicated results exceed those in the original papers on some datasets and metrics.

---

> > ### Comment · Reviewer_txHg · 2023-08-20
> >
> > Thank you for the responses. The responses have solved all my concerns. I am satisfied with it and would like to keep my previous scores (weak accept).

---

> > > ### Author Response · Authors · 2023-08-21
> > > **Response to Reviewer txHg**
> > >
> > > Your approval is very important to us and we will carefully revise our manuscript.

---

### Official Review · Reviewer_8GYm · 2023-07-03

**Soundness:** 3 good
**Presentation:** 3 good
**Contribution:** 3 good
**Rating:** 6
**Confidence:** 4

**Summary:**

This paper proposes a new framework for incomplete multi-view weak multi-label classification (iMvWMLC), which is a challenging task that involves missing views and labels in the data. The framework consists of four main components: a two-channel decoupling mechanism (called MTD) that extracts shared and view-specific features from each view, a cross-channel contrastive loss that enhances the semantic separation of the two channels, a random fragment masking strategy that reduces the redundancy of the input features, and a label-guided graph regularization loss that preserves the geometric structure among samples. The paper evaluates the framework on five datasets and compares it with eight state-of-the-art methods, showing that it outperforms them on various metrics and is robust to different levels of incompleteness.

**Strengths:**

(1) As far as I know, research on multi-view multi-label classification is still in its infancy, and this article focuses on this topic very well.

(2) To a certain extent, the logic and expression of this article are clear.

(3) I am interested in masking operations applied to vectorized feature data, most multimodal methods focus more on input real data such as text or images. I agree that the use of the mask mechanism on feature data is instructive.


**Weaknesses:**

(1) The similarity matrix calculated based on weak tags does not take into account the interference caused by unknown tags. The paper should address the potential interference caused by unknown tags in the calculation of the similarity matrix.

(2) The results of the ablation experiment suggest that the so-called contrast loss does not seem to have improved much, at least on the Corel5k dataset. The paper should provide some explanation.

(3) The paper needs to improve the details, as there is a error in section 3.4 where "TMD" is incorrectly referenced. The authors could correct this and prevent similar errors.

(4) The reviewer has noticed that there are some recent works aiming at solving the multi-view classification problem. The paper could do a better job by citing the different relevant works, e.g., [1-3], and clarifying the differences between this work and them.

[1] Trusted Multi-View Classification, ICLR’21

[2] Partially View-aligned Representation Learning with Noise-robust Contrastive Loss, CVPR’21

[3] Dual Contrastive Prediction for Incomplete Multi-View Representation Learning, TPAMI'23


**Questions:**

(1) It would be interesting to know if the authors have experimented with other masking methods, such as random bit masking of features, and what the results of such experiments were.

(2) Availability of code is important for reproducibility and further research. There is no code in the supplement material. Do the authors have any plans to make the code public?

(3) The paper should provide a more detailed discussion on why not all comparison methods were adapted to this task.


**Limitations:**

For missing multi-view data without a priori missing information, this method seems to be difficult to work.

---

> ### Author Rebuttal · Authors · 2023-08-10
>
> Your comments are greatly appreciated and we respond to these concerns as follows:
>
> W1: The similarity matrix calculated based on weak tags does not take into account the interference caused by unknown tags. The paper should address the potential interference caused by unknown tags in the calculation of the similarity matrix.
>
> A1: Thank you for your suggestion! In the process of calculating the similarity matrix, we only consider the sample similarity that can be obtained from the existing label information. However, in the calculation process of Eq. (4), unknown labels have an impact on the normalization scale, so we adjusted the calculation method of the similarity matrix to better handle sample labels with unknown tags. The specific adjustments are as follows:
>
> $T_{i,j} = \frac{C_{ij}\cdot(y_{i}y_{j}^T)}{C_{ij}\cdot(y_{i}y_{j}^T)+\eta}$, where ${T}\in [0,1]^{n\times n}$ is the sample similarity graph, describing the similarity between any two samples. $C_{ij} = G_{i,:}G_{j,:}^T$ is the number of label available for both samples $i$ and $j$. $y_{i}$ and $y_{j}$ denote the $i$-th and $j$-th rows of $Y$. And $\eta$ is a constant, empirically set to 100 for simplicity. Taking into account the fact that $C_{ij}$ is much larger than $y_{i}y_{j}^T$ on most datasets, when the number of categories is large, even if two samples only have one same label, it means that they are much more similar than other sample pairs.
>
> W2: The results of the ablation experiment suggest that the so-called contrast loss does not seem to have improved much, at least on the Corel5k dataset. The paper should provide some explanation.
>
> A2: In the ablation experiment, the contrastive loss did not show a large performance improvement on the corel5k dataset. We analyze that this may be because the cross-view sharing information of many samples in corel5k dataset is not sufficient, that is, the view-specific information occupies a dominant position, leading to difficulties in the aggregation of shared information. Experiments on other datasets show that the contrastive loss has improved in all metrics (Table 1 and Table 2 in new supplementary materials).
>
> W3: The paper needs to improve the details, as there is an error in section 3.4 where "TMD" is incorrectly referenced. The authors could correct this and prevent similar errors.
>
> A3: Thank you for your careful review. We have carefully checked the full text and corrected similar typos, and believe that the quality of the paper has been further improved.
>
> W4: The reviewer has noticed that there are some recent works aiming at solving the multi-view classification problem. The paper could do a better job by citing the different relevant works, e.g., [1-3], and clarifying the differences between this work and them.
>
> A4: Thanks! We have added to the manuscript an introduction to these multi-view classification works and the differences from these works in the paper:
>
> The trusted multi-view classification proposed by Han et al. focuses on the multi-view single-label classification based on confidence learning, trying to learn the classification uncertainty of each view based on evidence theory [1]. However, Dempster-Shafer evidence theory requires the mutual exclusion of labels, which is difficult to satisfy in multi-label classification. Yang et al. proposed a multi-view contrastive learning method with noise-robust loss to solve the partial view alignment problem, which takes aligned data as positive pairs and unaligned data as negative pairs [2]. Our method focuses on missing views rather than multi-view unalignment issue, and relies on the shared-private feature assumption rather than the prior alignment information for the construction of positive and negative pairs. Lin et al. proposed dual contrastive prediction for incomplete multi-view representation learning, which performs contrastive learning at both the instance level and category level to ensure cross-view consistency and recover missing information [3]. However, the instance-level contrastive loss of this method only aggregates cross-view representations in the potential space from the perspective of consistency, ignoring the view-private information.
>
>
>
> Q1: It would be interesting to know if the authors have experimented with other masking methods, such as random bit masking of features, and what the results of such experiments were.
>
> A1: Thanks for your question! In fact, we also tried to mask random bits of vector features, but it didn't work well, and we think that the continuity of key information in the features was badly broken.
>
> Q2: Availability of code is important for reproducibility and further research. There is no code in the supplement material. Do the authors have any plans to make the code public?
>
> A2: Yes, we will make the code public upon acceptance of the paper to help peers reproduce it.
>
> Q3: The paper should provide a more detailed discussion on why not all comparison methods were adapted to this task.
>
> A3: At present, the research on multi-view multi-label classification is in its infancy, and few methods that can handle both view and label incompleteness. In our comparison experiment, only iMvWLL[4], NAIM3L[5], and DICNet[6] can meet our task setting at the same time. Therefore, following the existing methods [4-6], we introduce other methods that can handle incomplete multi-views or partial multi-labels in the experiment to enrich our experiment, which also can demonstrate the necessity of special design for missing views and unknown labels in the network structure.
>
> [4] Incomplete multi-view weak-label learning, IJCAI
>
> [5] A concise yet effective model for non-aligned incomplete multi-view and missing multi-label learning, TPAMI
>
> [6] DICNet: Deep Instance-Level Contrastive Network for Double Incomplete Multi-View Multi-Label Classification, AAAI

---

> > ### Comment · Reviewer_8GYm · 2023-08-14
> >
> > Thanks for the authors' responses. The authors have addressed my concerns and I will maintain my rating of acceptance for the paper.

---

> > > ### Author Response · Authors · 2023-08-21
> > > **Response to Reviewer 8GYm**
> > >
> > > Thank you for your approbation and we will carefully integrate your suggestions into our manuscript.

---

### Official Review · Reviewer_NrND · 2023-07-12

**Soundness:** 2 fair
**Presentation:** 2 fair
**Contribution:** 2 fair
**Rating:** 4
**Confidence:** 4

**Summary:**

This paper studies incomplete multi-view weak multi-label learning problem, which is important. The authors propose a masked two-channel decoupling framework based on deep neural networks. They develop cross-channel contrastive loss, a label- guided graph regularization loss, and random fragment masking strategy. The experiments validate the effectiveness.

**Strengths:**

- The problem is interesting.
- The techniques in this work seem correct and reasonable.

**Weaknesses:**

- This work just combines some existing widely-used techniques. Thus the work is a bit incremental, does not provide new insights to me.
- As stated in abstract, the core is to decouple view-level representation into  shared representation and a view-proprietary representation. This idea has been widely used in existing works.
- The authors first fill the missing views with noise, and missing label with 0, and then treat it as normal complete view learning problem. To me, this scheme seems not grasp the essence of multi-view weak multi-label learning problem.
- The aim and construction of contrastive loss in this work seem a bit confusing to me. It is suggested to explain it in more detail.

**Questions:**

See weakness part

**Limitations:**

The author seems to lack a comprehensive discussion on the limitations of the proposed method.

---

> ### Author Rebuttal · Authors · 2023-08-09
>
> Thank you for your efforts in reviewing the manuscript! We respond to the comments below:
>
> W1. This work just combines some existing widely-used techniques. Thus the work is a bit incremental, does not provide new insights to me.
>
> A1. Indeed, we admit that the techniques or ideas used in our method have similar applications in other fields. However, we emphasize more on the combination of these existing ideas and multi-view multi-label classification tasks, that is, solving the problem “Why should them be used on the MvMLC task, How to use them, and What are the benefits?”.
>
> The following is the difference and significance of the three core technologies involved in this paper and the existing technologies:
>
> 1)	View-specific representation decoupling framework and contrastive loss. Decoupling the features of each view or modality into a shared feature and a proprietary feature is not new in multi-view or multi-modal learning [1,2], however, the crux of the problem is how to achieve this decoupled state, the technical routes and emphases of existing methods are not the same. For instances, work [1] uses an adversarial loss to make it difficult for the model to discern the view origin of shared features, however, the training cost of adversarial learning is high. Our decoupling framework and corresponding contrastive loss avoid this problem well. Work [2] only focuses on reducing the distance between shared feature pairs and increasing the distance between private feature pairs, ignoring the differences between private features and shared features at both the intra-view and inter-view levels. However, our method considers both: maximizing the distance between shared and private features from individual view-specific feature, and the comparison between shared and private feature sets from different views.
> 2)	Label-guided graph regularization. The Laplacian graph regularization technique is often used in traditional methods in the field of multi-view learning, and at the core of the technique, the construction of Laplacian graph usually only comes from the original feature space and acts on the latent representation of the sample. In our MvMLC task, we boldly combine it with the characteristics of the multi-label task, that is, construct a label-based sample adjacency matrix to replace the adjacency matrix from the original data. The obvious advantage of this design is that the sample label information has clear semantic information while the original data often contains noise and redundancy.
> 3)	Masking random fragment of feature. As another work we are proud of, masking attempts on random segments is our contribution to the multi-view learning community. As far as we know, masking strategy that is currently popular in fields such as images have not been applied to vector data, especially in the field of multi-view or multi-label. And unlike the well-known MAE technology, we do not need to perform additional non-coding processing on the masked data segments, which can be easily transplanted to any multi-view learning network to a large extent, and it is a kind of Plug and play technology.
>
> [1] Multi-View Multi-Label Learning with View-Specific Information Extraction. IJCAI,2019.
> [2] CDDFuse: Correlation-Driven Dual-Branch Feature Decomposition for Multi-Modality Image Fusion. CVPR,2022.
>
> W2. As stated in abstract, the core is to decouple view-level representation into shared representation and a view-proprietary representation. This idea has been widely used in existing works.
>
> A2. Thank you for the comment. As we mentioned in A1, some methods apply the idea of decoupling in multi-view learning. However, the above purpose cannot be achieved by using only two types of encoders. And the key to this idea is to design a suitable strategy to make the features extracted by the two types of encoders meet our assumption, that is, true shared features and proprietary features. Our experiment in Figure 3 also confirms the effectiveness of our strategy.
>
> W3. The authors first fill the missing views with noise, and missing label with 0, and then treat it as normal complete view learning problem. To me, this scheme seems not grasp the essence of multi-view weak multi-label learning problem.
>
> A3. Indeed, we fill missing instances with noise and fill unknown labels with 0 values, however this is just a means of data preprocessing. And the purpose of that is to ensure that multi-view data is dimensionally aligned, which is very important in batch-based training of deep neural networks. It should be noted that although we fill in the missing information beforehand, we do not treat it as complete multi-view learning in subsequent processing. For example, in Eq. (1), we introduce condition $\Upsilon$ to avoid filled noise from taking part in the calculation of contrastive loss; In Eq. (2) and (3), we introduce missing index matrix $W$ in the process of multi-view fusion to exclude the illegal features corresponding to missing instances. In Eq. (9), we introduce a missing label indicator matrix $G$ to avoid the negative effects of unknown labels.
>
> On the other hand, we would like to say that there are many ways to deal with missing views and labels, including missing view recovery, pseudo-label filling, etc., and the "skip" strategy adopted by our method based on prior missing information is also used by many related methods [3-5].
>
> [3] A Concise yet Effective Model for Non-Aligned Incomplete Multi-view and Missing Multi-label Learning. TPAMI,2022.
> [4] Dicnet: Deep instance-level contrastive network for double incomplete multi-view multi-label classification. AAAI, 2023.
> [5] Expand globally, shrink locally: Discriminant multi-label learning with missing labels, Pattern Recognition, 2021.
>
> Responses to W4 will be added to the comments section after the rebuttal phase begins due to word limitation.

---

> > ### Author Response · Authors · 2023-08-10
> > **Response to weakness 4**
> >
> > W4. The aim and construction of contrastive loss in this work seem a bit confusing to me. It is suggested to explain it in more detail.
> >
> > A4. Thanks for the comment, and we are sorry to confuse you about the contrastive loss. According to your suggestion, we further explain the design idea of contrastive loss as follows in the manuscript:
> >
> > For any sample $i$, we pair features on $2N$ channels in pairs ($4N^2$ feature pairs in total), and classify these feature pairs into 2 categories, namely positive pairs and negative pairs, where positive pairs consist of shared features from different views, and the remaining “shared-private” and “private-private” feature pairs are negative pairs. Our goal is to minimize the distance between positive instance pairs (the denominator part), while maximizing the distance between negative pairs (the numerator part). Here we not only consider maximizing the difference between private features decoupled from the different view ($o_{i}^{(u)}$ and $o_{i}^{(v)}$, $u\neq v$), but also reduce the similarity of shared features and private features from both intra-view and inter-view levels ($s_{i}^{(u)}$ and $o_{i}^{(v)}$, including $u=v$). This approach fulfills the design objective of the dual-channel model, which is to encourage consistency between shared features from different views while maintaining a clear distinction between view-proprietary feature and shared or other view-proprietary features.
> >
> > In addition, considering the efficiency of matrix calculation, we set the coefficient of the contrastive item of shared features and private features to 2 in the numerator part, so that the entire paired similarities can be obtained by a single matrix multiplication, i.e., $[s_i^{(1)}; s_i^{(2)};…; s_i^{(N)}; o_i^{(1)}; o_i^{(2)};…; o_i^{(N)}]\times [s_i^{(1)}; s_i^{(2)};…; s_i^{(N)}; o_i^{(1)}; o_i^{(2)};…; o_i^{(N)}]=Q\in \mathbb{R}^{2N\times 2N}$. The matrix $Q$ can be divided into four sub-matrices whose size is $N$ by $N$. The sub-matrix in the upper left corner stores the similarity values of the denominator in Eq. (1), the sub-matrix in the lower right corner stores the values of the second item in the numerator, and the remaining two submatrix, which are transposed to each other, store the similarity values of the first item in the numerator. Therefore, we set the coefficient of the first term in the numerator to 2. At the same time, the diagonal elements of $Q$ are removed, so that the number of elements in the numerator and denominator parts is $\frac{1}{3N^{2}-N}$ and $\frac{1}{N^{2}-N}$ respectively.
> >
> > About limitation of the paper:
> >
> > In the conclusion, we explain the limitations of our method and further improvement directions in the future, such as adding the consideration of label correlation to the multi-label classification problem in the model. At the same time, for missing views and partial multi-label, in addition to introducing prior information, more complex view completion and pseudo-label prediction can be performed to improve the discriminability of the model.

---

> > > ### Comment · Reviewer_NrND · 2023-08-22
> > >
> > > Thank you for your feedback. The authors have addressed some of my concerns and I will increase my score.

---

> > > > ### Author Response · Authors · 2023-08-22
> > > > **Response to Reviewer NrND**
> > > >
> > > > Thank you very much for your response and we would be happy to clarify further if you have additional concerns.

---

### Author Rebuttal · Authors · 2023-08-10

Thanks to all reviewers, and this is our new supplementary material.

---

### Decision · Program_Chairs · 2023-09-21

**Decision:**

Accept (poster)

**Comment:**

This paper focuses on the complex yet highly realistic task of incomplete multi-view weak multi-label learning, and proposes a masked two-channel decoupling framework based on deep neural networks. The method decouples the single-channel view-level representation into a shared representation and a view-proprietary representation, and designs a cross-channel contrastive loss to enhance the semantic property of the two channels. After full discussion, the review issues were resolved, and the method was deemed to be innovative, well-experimented, and exactly described. I'm willing to accept manuscript.